# Using Architectural Mapping to Understand Behavior and Space Utilization in a Surgical Waiting Room of a Safety Net Hospital

**DOI:** 10.3390/ijerph192113870

**Published:** 2022-10-25

**Authors:** Elizabeth N. Liao, Lara Z. Chehab, Michelle Ossmann, Benjamin Alpers, Devika Patel, Amanda Sammann

**Affiliations:** 1Division of General Surgery, Department of General Surgery, University of California (San Francisco), San Francisco, CA 94143, USA; 2Global Research & Insights, MillerKnoll, Zeeland, MI 49464, USA

**Keywords:** heat mapping, waiting experience, patient behavior, design, healthcare

## Abstract

Objective: To use architectural mapping to understand how patients and families utilize the waiting space at an outpatient surgery clinic in a safety-net hospital. Background: The waiting period is an important component of patient experience and satisfaction. Studies have found that patients value privacy, information transparency and comfort. However, approaches common in the architecture field have rarely been used to investigate interactions between patients and the built environment in a safety-net healthcare setting. Methods: This was a prospective observational study in a general surgery outpatient clinic at a safety-net hospital and level 1 trauma center. We used a web-based application generated from the design and architecture industry, to quantitatively track waiting space utilization over 2 months. Results: A total of 728 observations were recorded across 5 variables: time, location, chair selection, person/object, and activity. There were 536 (74%) observations involving people and 179 (25%) involving personal items. People most frequently occupied chairs facing the door (43%, *n* = 211), and least frequently occupied seats in the hallway (5%, *n* = 23), regardless of the time of their appointment (*p*-value = 0.92). Most common activities included interacting with personal phone, gazing into space, and talking face to face. Thirteen percent of people brought mobility devices, and 64% of objects were placed on an adjacent chair, indicating the desire for increased personal space. Conclusion: Architectural behavioral mapping is an effective information gathering tool to help design waiting space improvement in the safety-net healthcare setting.

## 1. Introduction

Healthcare waiting spaces play an important role in the patient experience. Patients often arrive to clinics and hospitals feeling anxious and stressed, thereby increasing sensitivity not just to the clinical outcome, but also to their experience throughout their visit [1,2]. It has been shown that waiting spaces have the power to quell or exacerbate these feelings of anxiety, and to increase or decrease trust among patients for their healthcare providers [1,3]. As such, the waiting experience significantly contributes to patients’ perception of the quality of care they receive [4]. Unfortunately due to limited resources and longer wait times at safety-net hospitals and community clinics [5], the waiting experience is not often prioritized for publicly insured patients. Medicaid- and un-insured patients systematically experience disparities in care compared to their privately-insured counterparts, including exposure to extended office wait times and poorly designed or outdated physical spaces [6]. In our previous study investigating patient flow in an outpatient surgery clinic situated in a safety-net hospital, we found that patients spent an average of 25.4% of their time in the waiting space before they were called into an exam room, and an additional 43.0% of their time sitting idle without a provider in the exam room [7].

Poorly designed waiting spaces can increase barriers to accessing healthcare. Patients cite unpleasant waiting spaces as one of the leading reasons for not returning to a healthcare facility [8], describing them as “cramped holding pens” [9]. In addition, how patients decide whether to wait or to leave without being seen has been shown to be partially informed by the waiting experience [10,11,12]. For instance, waiting spaces that created activities for patients to use the waiting time constructively increased patients’ willingness to wait [10].

Well-designed waiting spaces, on the other hand, have been shown to improve perceived quality of care [4,13,14] and medical literacy [15,16,17,18,19], as well as help to alleviate physical pain and emotional stress through the engagement of the senses [20,21,22,23]. As such, innovation around the waiting space has centered around decreasing wait times and/or improving time perception [24,25,26,27], introducing positive distractors such as televisions and artwork [28,29,30,31,32], and improving patient privacy, information transparency, and comfort [33,34]. Designing waiting spaces with the patient at the center helps patients feel valued and increases patient perceptions on quality of healthcare delivery [4].

Behavioral mapping is an approach to understanding how physical space is utilized and how the environment can influence certain behaviors. This approach requires few resources and can be achieved through use of a paper and pen or a computer-based system [35]. By recording activities on floor plans, diagrams, or maps [35,36,37], researchers can better understand the effect of the built environment on a target population, and subsequently design person-centered solutions to physical spaces. While behavioral mapping is ubiquitous in the field of architecture, few studies have used behavioral mapping in health experience design research. One study used a combination of paper-based collection forms and unstructured observational data to show how the layout of the waiting space influenced patient flow and congestion in 3 large general hospitals in China. Investigators found that water fountains, while provided inside the hospital, were not optimized for patient and family access because they were too far away for parents in the waiting room to make infant formula [38]. Another study used virtual reality to show that participants considered factors such as seat comfort, visual and auditory privacy, and access to information when choosing a seat while waiting and that the priority of these factors shifted depending on their reason for waiting [39]. Only a few studies have focused on waiting spaces for vulnerable populations in specialty ambulatory care settings, despite the longer wait times and lower satisfaction rates [40,41,42,43,44]. To our knowledge, no studies have examined the patient waiting experience in outpatient hospital ambulatory care clinics in an underserved population [45]. There is an essential need to understand how the physical environment is utilized in order to produce patient-centered solutions that improve the waiting experience for the Medicaid- and un-insured populations.

In this exploratory study, we used behavioral mapping to better understand who frequented the waiting space (patients, families, caregivers, etc.), how visitors chose to spend their time, and how they utilized the physical space. By developing a data-driven understanding of visitors and their behaviors and preferences, we sought to gain insights that could inform patient and family-centered solutions to improving access and experience for underserved populations seeking specialty care.

## 2. Materials and Methods

### 2.1. Setting

We conducted a prospective observational study in a general surgery outpatient clinic at a safety-net teaching hospital and level 1 trauma center affiliated with an academic medical center. Between 2008 and 2018, the clinic saw 17,632 patients across 65,211 visits. Typical visit types include new patient consults from other specialties, pre-operation surgery patients and post-operation follow-up patients. The average age was 52.5 years (SD 15.3), 64.7% identified as male, 74.2% did not identify as Non-Hispanic White, and 33.6% were non-English speakers [7] (Table 1).

At the time of this study, each clinic was staffed by a general surgeon, a nurse practitioner and two medical assistants. Since the clinic took place in an academic teaching hospital, medical students attended irregularly, depending on didactic and inpatient clinical activities. There were no residents in the clinic. Morning clinics were held between 9 a.m. and 12 p.m., and afternoon clinics were held between 1 p.m. and 4 p.m. There were 4 scheduled attending-led clinics per week (3 morning and 1 afternoon) and 1 scheduled nurse-practitioner-led clinic per week (afternoon). All study activities were conducted in the patient waiting space in the clinic. This study was approved by the UCSF IRB as an expedited protocol. Verbal consent was approved for this study as the study presented no more than minimal risk of harm to subjects and involved no procedures.

### 2.2. Description of the Waiting Space

The outpatient surgery waiting space contained 16 chairs available to patients and their families: 12 chairs that were lined along the perimeter of a designated waiting room, and 4 chairs in the hallway. Photographs and the floorplan with the chairs labeled from 1–16 are shown in Figure 1. 

The waiting room was a rectangular shape, with all but one short wall containing chairs. There was no other furniture available to patients and families in the waiting spaces. The hospital is equipped free Wi-Fi. As noted in Figure 1: wall E contained a television, some artwork and a trash bin; wall F contained 5 chairs and a set of four large windows; wall B contained 3 chairs and posted announcements of events and health education material; wall C contained 4 chairs, posted announcements of events in the hospital, a clock, and a door that was permanently locked; and wall G contained 4 chairs that were just outside of the waiting room in the hallway leading to the clinic’s exam rooms. Chairs on wall F will hereafter be referred to as “chairs facing the door”; chairs on wall B will hereafter be referred to as “chairs facing the TV”; chairs on wall C will hereafter be referred to as “chairs facing the windows”; and chairs on wall G will hereafter be referred to as “chairs in the hallway”. Patients who sit in the chairs facing the window have views of hospital buildings and trees.

### 2.3. Architectural Mapping

We used a computer-based approach to behavioral mapping. DOTT is a web-based application created by EwingCole, PA and Interior Architecture & Design, PLLC [46]. We chose to use this tool due to its ease of use and its ability to systematically document a larger number of entries across multiple variables over time. Data were collected over a 2-month period from June 1 to July 31, 2019. A team of 4 researchers performed a total of 24 h of observations across 11 clinics. For each clinic period, we tracked the following variables every 10 min: “occupant type” (patient, child/family member, caretaker, etc.), “seat occupancy” (chairs 1–12), “activity” (reading, talking on phone, interacting with phone, etc.), and “personal items” (stroller, crutches/wheelchair, bags, etc.) We chose to track these variables in a 10-min time interval based on our previous time-tracking study [7] where we found that patients waited an average of 24.4 min (range: 0–110 min) in order to balance documenting new arrivals and changes in activity and patterns in seat occupancy.

### 2.4. Data Analysis

The data from DOTT were exported into an Excel spreadsheet, then cleaned and standardized by author ENL: each object and each person were logged as separate entries if they were once one entry. For example, a person nursing a baby was re-organized into two entries (two people). Free text notes were translated into categories (e.g., “person holding purse sitting in chair” became “person and personal item in chair”.) Frequencies were calculated for each categorical variable, and binomial variables were compared with two-tailed Fisher’s exact test. All statistical tests were completed in Stata, version 17. Figures were the authors’ own elaboration and made in a variety of programs, including Canva and Notability.

## 3. Results

### 3.1. Description of Observations

A total of 728 observations were recorded in DOTT, 74% of which involved people, 25% involved personal items, and 2% were entries of an empty room (Table 2).

Afternoon clinics were slightly more busy than morning clinics: 52% of observations were recorded during the 6 morning clinics and 42% were recorded during the 4 afternoon clinics. Five percent of observations had erroneous timestamps and were excluded from the dataset (times were incorrectly generated DOTT, e.g., stating that clinic took place at 9 p.m.) Families were 23% more likely to be present in morning clinics than in afternoon clinics (*p* = 0.001). Although not statistically significant, there was no difference between presence of mobility devices in morning versus afternoon clinics (*p* = 0.22).

### 3.2. Patterns in Seat Occupancy: Direction of Seating

Ninety-two percent of people chose to sit in one of the 16 chairs in the waiting space. The remainder sat in their own wheelchair or baby stroller (6%), remained standing (1%), or walked around (1%). On average, the chairs facing the door were most frequently occupied (43.4%), followed by the chairs facing the TV (26.5%) and chairs facing the window (25.4%.) In contrast, chairs in the hallway were only occupied in 1.8% of observations

### 3.3. Patterns in Seat Occupancy: Chair Selection

Overall, the seats facing the door were most frequently occupied, whereas the seats in the hallway were the least occupied. See Figure 2 for a visual heat map demonstrating patterns in seat occupancy, and Table 3 for observed frequencies organized from highest to lowest and Table 4 for observed frequencies for patients and family members.

### 3.4. Patterns in Seat Occupancy: Time of Day

Although not statistically significant, seat preferences did not differ between morning and afternoon clinics (*p* = 0.92). See Table 5 for frequency of chair occupancy in morning and afternoon clinics. Overall, the chairs facing the door were occupied the most and the chairs in the hallway were occupied the least.

### 3.5. Personal Items

Of the 728 observations recorded in DOTT, 179 were personal items: 40% were mobility devices (including wheelchairs, strollers, front-wheel walkers and canes), 40% were purses or bags, and 20% were miscellaneous items that could not fit in the individual’s pocket such as food and clothing. 

As shown in Figure 3, if a person brought a personal item, they placed it on their lap or on an adjacent chair(s); no one placed their personal item on the ground. Placement preferences differed depending on where people sat. People who sat in chairs facing the door most frequently chose to place their item in an adjacent chair, whereas people sitting in chairs facing the window placed their items on their laps or an adjacent chair almost equally. 

### 3.6. Activities

Once patients and their family members entered the waiting space, they spent their time doing between zero to three activities: 16% people had zero activities recorded, 77% had 1 activity, 7% had 2 activities, and (1%) had 3 activities. The activities were categorized into interacting with a personal item or person brought into the clinic (56%) or interacting with the physical environment (44%) (Figure 4). By far, the most common activity was interacting with a personal phone.

## 4. Discussion

Safety-net hospitals are low-resourced, space-constrained and often in older buildings not equipped for modern needs, such as electric wheelchairs, WIFI, and phone charger stations. Furthermore, these buildings may not be easily amenable to structural modifications. Historically, space improvement has not been high on the priority list for these hospitals and has generally meant updating furnishings, such as buying new chairs and updating the decor. These solutions, unfortunately, are largely based on the general population, rather than with the Medicaid population [4,30,47]. We chose to use behavioral mapping to help us build tailored, resource-appropriate solutions for underserved populations. Three key findings emerged about how the waiting space at a safety-net surgical clinic was utilized in terms of occupant type, location, time, and activities. First, 43% of observed patients and their families preferred to sit in a seat facing the door, rather than oriented towards the TV or window. Second, 33% of patients brought a larger object requiring an additional space for placement, such as a mobility device or a purse, and patients occupying seats facing the door preferred to place their personal item on an adjacent chair (64%), while patients occupying other seats in the clinic had no strong preference. Finally, individuals who engaged in an activity chose to interact with an object or a person they personally brought into the clinic (56%) or chose to interact with the physical environment as they waited to be called by staff (44%).

Our findings support previous literature in showing that patients value information transparency, privacy, and relaxing environments [1,13,33,48]. For instance, one study found that people prefer not to wait in the hallway due to minimal privacy [49]. Another reported that patients value information transparency and relaxing environments [50]. Similarly, we posit that patients and families disproportionately select the seats facing the door inside the waiting room to balance their desire for information transparency and privacy. Seats facing the door have much higher visual access to the ongoings of clinic operations than other seats in the waiting space, suggesting that patients choose those seats to maximize access to information about when they might be called for their appointment. However, patients and families also value privacy, as they prioritize seats in the waiting room versus in the hallway—the latter of which have the highest level of information access but the lowest level of personal space and privacy. Future design in this space requires balancing information transparency, privacy, and relaxing environments to improve the patient waiting experience.

Our findings support prior research demonstrating that patients and families may use their personal items as a way of creating and expanding their personal space and erecting privacy barriers [33,51,52], as over half of observed patients placed a larger personal item on an adjacent chair, preventing others from occupying the chair next to them. Future space design and furniture selection should consider enabling patients to create privacy and/or physical barriers. For those who currently place their belongings on an adjacent chair due to lack of space, creating space for people to store their belongings, especially mobility devices, could be a low-cost solution to improve accessibility.

Previous literature has also shown that patients value environmental distractions and nature views to decrease stress and anxiety [28,47]. In our study, patients did not show a preference for chairs facing the window and only 8% of observations involved looking out the window. Instead, the most common activity (38%) was interacting with phones. Investigating what patients and their families value in their environmental distractions and optimizing use of technology in these spaces is key to improving waiting experiences.

There are several re-organization potentials based on our findings. First, chairs can be reoriented so patients face the door to increase information transparency (Figure 5B), however, workflow solutions aimed at increasing transparency and reducing wait times would produce a more sustainable solution to this problem. Second, the waiting room could be filled with sitting nooks that service both groups and individuals as a way to create a sense of privacy and personal space (Figure 5C) Third, providing different seating options for visitors with mobility restrictions (Figure 5D).

We faced several limitations related to the technology and data collection used in this study. First, DOTT cannot track people or objects over time as it is designed to document information every 10 min. If a person sat in the same seat for 40 min, it would appear as four separate entries in the dataset. Similarly, a seat that is occupied by a new person every 10 min over 40 min would also appear as four separate entries in the dataset. As such, we could not determine the sequence of events leading to which seat(s) are the most popular if the waiting room was empty, which seat(s) are popular if one chair is occupied, etc. It is also not always possible to determine whether a person placed their person item on their right or left chair. Still, we can identify patterns regarding which seats are being chosen and what people do with their time. Second, 15% of observations involving people (*n* = 84) had zero activities recorded. This indicates a recording error as all people do something when they are waiting. Most likely, these people were gazing into space at that time (hence having “zero” activity). We tried to decrease recording error by standardizing classifications of all categorical data by encouraging the use of the free-text section of DOTT in case a person, object, and/or activity could not easily be classified in the existing system.

Hospitals and clinics are meant to be healing spaces; buildings are vehicles for transparency, healthcare access and delivery. Establishing best practices for spaces and understanding the different contexts and needs in which these spaces are used requires a multi-disciplinary team. Historically, this team has consisted of architects and engineers, with minimal input from providers and patients. It is important for providers to be part of this process, bringing their expertise in patient care to maximize the ways we use elements of space design, such as colors, noise, light, and sound to help the healing process and decrease barriers to healthcare access.

## 5. Conclusions

Using a web-based behavioral mapping tool to understand patient behaviors, we obtained pilot data about how and where people spend their time in a minimally furnished surgical clinic waiting space. Our findings showed how the waiting space at a safety-net surgical clinic was utilized in terms of occupant type, location, time, and activities. Our next steps are to prototype different seating arrangements for visitors, and iterate on solutions based on patient and family behavioral mapping data and qualitative feedback. By demonstrating the use of this low-resource behavioral mapping tool at our safety-net hospital, we hope others will find this approach useful in investigating and improving their waiting spaces.

## Figures and Tables

**Figure 1 ijerph-19-13870-f001:**
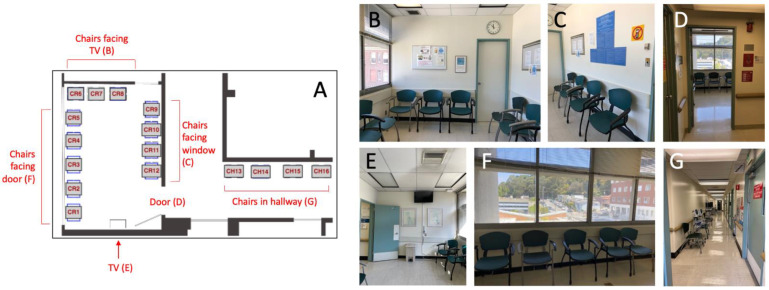
Floorplan of the waiting space with corresponding photos. (**A**) Floorplan labeled with chair types, TV, and door. (**B**) Photo of the chairs facing the TV (chairs 6–8). (**C**) Photo of the chairs facing the window (chairs 9–12). These chairs line the same wall as the door. (**D**) Photo of entrance of the waiting room from the hallway; (**E**) Photo of wall with TV. (**F**) Photo of the chairs facing the door (chairs 1–5). These face the doorway and have their backs to the windows. (**G**) Photo of the chairs in the hallway (chairs 13–16).

**Figure 2 ijerph-19-13870-f002:**
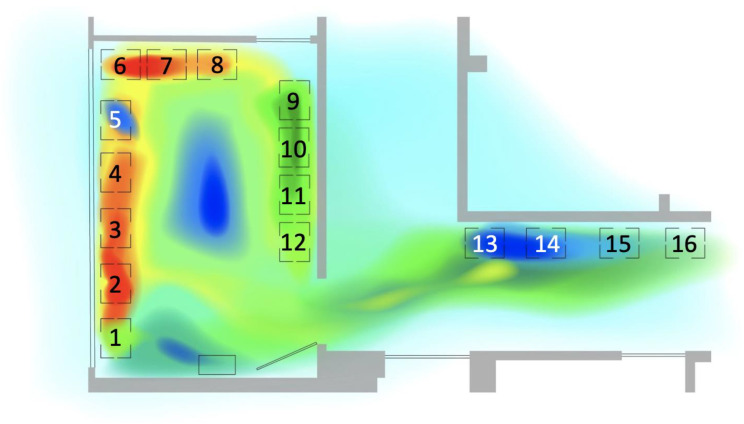
Heat map of locations of people. DOTT recorded people’s locations as frequencies of their X and Y coordinates. These frequencies were used to map ‘hot’ (most frequently occupied, indicated by the warm colors such as red and orange) and ‘cool’ (least frequently occupied, indicated by the cool colors, such as blue and green) spots across the waiting space. The numbers 1–16 refer to chairs 1–16.

**Figure 3 ijerph-19-13870-f003:**
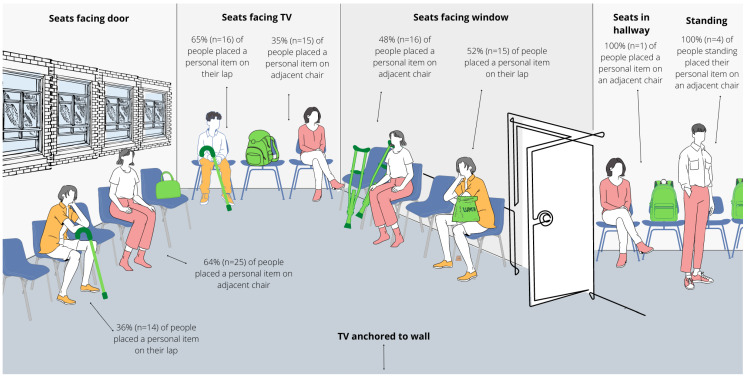
Infographic of how people chose to place their personal items.

**Figure 4 ijerph-19-13870-f004:**
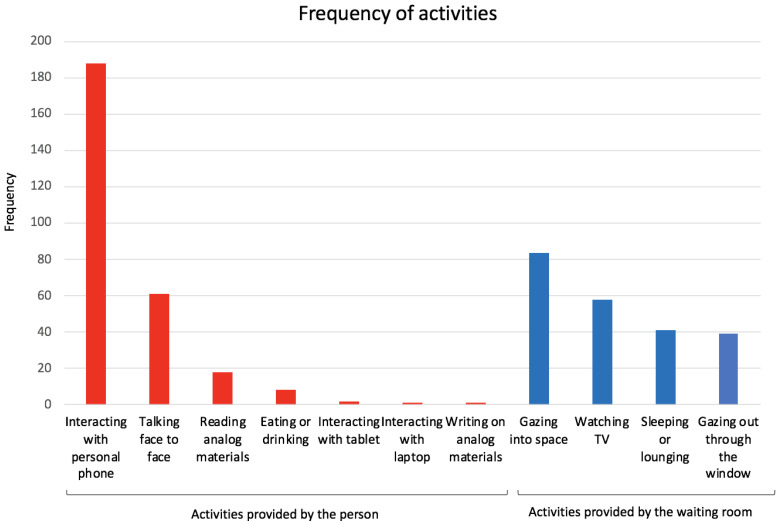
Frequency of activities.

**Figure 5 ijerph-19-13870-f005:**
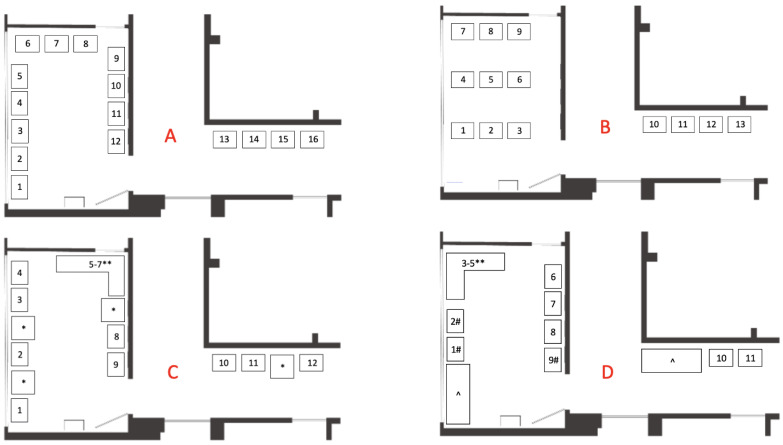
Potential re-organized floorplans. The numbers refer to chairs. (**A**) Current floorplan. (**B**) Potential floorplan that prioritizes information transparency. (**C**) Potential floorplan that prioritizes privacy and personal space. *: tables; **: sofa that can sit groups of three. (**D**) Potential floorplan that prioritizes providing different seting options for visitors with mobility restrictions. ^: open spots for wheel-chair users and strollers; #: specialized chairs that help patients stand up (e.g., those who have hernias and/or use crutches or canes); **: sofa that can sit groups of three.

**Table 1 ijerph-19-13870-t001:** Sociodemographics of patients seen in general surgery outpatient clinic from 2008 to 2018.

Demographics	Patients Seen (*n* = 17,632) No. (%)
Age, average (standard deviation), years	52.5 (15.3)
Gender	
- Female	22,614 (34.8)
- Male	11,412 (64.7)
- Unknown	221 (0.3)
Race/Ethnicity	
- Hispanic	1153 (6.5)
- Non-Hispanic Black	3053 (17.3)
- Non-Hispanic Asian	3285 (18.6)
- Non-Hispanic White	4535 (25.7)
- Non-Hispanic Other	5280 (30.0)
- Unknown or Decline to States	326 (1.8)
Primary language	
- English	11,695 (66.3)
- Spanish	3303 (18.7)
- Cantonese	1320 (7.5)
- Other language	1166 (6.6)
- Unknown	148 (0.84)
Family size	
- One member	14,305 (81.1)
- Two members	1807 (10.3)
- Three members	678 (3.9)
- Four members	521 (3.0)
- Greater than four members	321 (1.8)
Income source	
- Professional/technical *	343 (2.0)
- Labor/production ^	1144 (6.5)
- Service/sales	2148 (12.2)
- Retirement income	336 (1.9)
- Disability income	822 (4.7)
- General or public assistance	1023 (5.8)
- Other **	1049 (6.0)
- None	10,767 (61.1)

* This is inclusive of executive, administrative, managerial, professional, technical and related support); ^ This is inclusive of production, inspection, repair, craft, handlers, helpers, labors, and transportation; ** This is inclusive of Veteran Affairs Benefits, interest, dividends, rent, child support, alimony, etc.

**Table 2 ijerph-19-13870-t002:** Overview of DOTT observations.

Description	Frequency *n* (%)
Observations Involving People (*n* = 536)	
Patient	408 (76)
Family Members	125 (23)
Staff Members	3 (1)
Observations Involving Personal Items (*n* = 179)	
Mobility related items (wheelchair, crutches, cast, walker, baby stroller)	72 (40)
Purse or bag	71 (40)
Miscellaneous (e.g., food, clothing)	36 (2)

**Table 3 ijerph-19-13870-t003:** Frequency of Chair Selection organized from highest to lowest.

Seat Orientation	Seat Number	Occupation Frequency, % (*n*)
Facing the door	2	10.9% (53)
Facing the door	3	10.5% (51)
Facing the door	4	9.7% (47)
Facing the TV	6	9.7% (47)
Facing the TV	7	9.0% (44)
Facing the door	1	8.4% (41)
Facing the TV	8	7.8% (38)
Facing the window	12	7.8% (38)
Facing the window	11	6.2% (30)
Facing the window	9	5.7% (28)
Facing the window	10	5.7% (28)
Facing the door	5	3.9% (19)
In the hallway	13	2.9% (14)
In the hallway	14	1.0% (5)
In the hallway	15	0.8% (4)
In the hallway	16	0% (0)

**Table 4 ijerph-19-13870-t004:** Frequency of Chair Selection for patients and family members.

Seat Number	Seat Orientation	Occupation Frequency by Patients (%, *n*)	Occupation Frequency by Family Members, % (*n*)
1	Facing the door	7.8% (29)	10.5% (12)
2	Facing the door	10.8% (40)	11.4% (13)
3	Facing the door	10.2% (38)	11.4% (13)
4	Facing the door	10.0% (37)	8.8% (10)
5	Facing the door	3.8% (14)	4.4% (5)
6	Facing the TV	10.5% (39)	7.0% (8)
7	Facing the TV	10.0% (37)	6.1% (7)
8	Facing the TV	6.2% (23)	12.3% (14)
9	Facing the window	7.6% (28)	0% (0)
10	Facing the window	5.7% (21)	7.1% (7)
11	Facing the window	5.4% (20)	8.8% (10)
12	Facing the window	8.1% (30)	7.0% (8)
13	In the hallway	2.4% (9)	3.5% (4)
14	In the hallway	0.5% (2)	1.8% (2)
15	In the hallway	1.1% (4)	1.0% (1)
16	In the hallway	0% (0)	0% (0)

**Table 5 ijerph-19-13870-t005:** Chair occupancy in morning and afternoon clinics.

Chair Type	Morning Clinic *n* (%)	Afternoon Clinic *n* (%)
Chairs facing the door	119 (45)	88 (45)
Chairs facing the TV	69 (26)	50 (25)
Chairs facing the window	69 (26)	48 (23)
Chairs facing the hallway	8 (3)	14 (7)

## Data Availability

The datasets used and/or analyzed in this study are available from the corresponding author on reasonable request.

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
