# Peer review of "Using Architectural Mapping to Understand Behavior and Space Utilization in a Surgical Waiting Room of a Safety Net Hospital"

_ijerph, 2022, doi:10.3390/ijerph192113870_

Round 1
Reviewer 1 Report
|
Paper Review |
|
|
|
Title: “Using Architectural Mapping to Understand Behavior and Space Utilization in a Surgical Waiting Room of a Safety Net Hospital” |
|
Authors: Elizabeth N. Liao, Lara Z. Chehab, Michelle Ossmann, Benjamin Alpers, Devika Patel and Amanda Sammann |
|
Manuscript ID: IJERPH–1960753 |
|
Journal: Int. J. Environ. Res. Public Health |
|
|
|
Review Comments
This paper studies the behavior and space utilization of patients at an outpatient surgery clinic in a safety net hospital. There are some critical issues and concerns, and the authors should address the following comments.
|
|
Comment 1. The English of the paper needs to be improved. There are some syntax and grammar errors. |
|
Comment 2. Citation of references in the text is out of order. Please do a revision. |
|
Comment 3. In line 23, what does “(p=0.92)” mean? Describe it in the text. |
|
Comment 4. In line 35, there is a superscript of “1” for the word “anxiety”. What is it for? Please revise. |
|
Comment 5. In data gathering using DOTT, did you consider changes in patient’s behavior? For example, change in place? Or, change in items they used while waiting? |
|
Comment 6. Please present a spectrum (range of colors according to using frequency) for Fig. 2. For example, from Red color for most–used to Blue color for least–used. |
|
Comment 7. In Subsection “3.5. Personal items”, do you know whether people place their items on their right or left chair? |
|
Comment 8. Please determine whether the center provides free Wi–Fi connection or not? Is the waiting room equipped with free Wi–Fi connection or not? |
|
Comment 9. In Abstract, please bold the words Objective, Background, Methods, Results, Conclusion to make them more distinguishable. |
|
Best Regards, |
|
|
Author Response
- These have been edited where needed.
- Thanks, the last 4 citations were re-arranged. The citations appear in order of appearance in the manuscript.
- P refers to the “p-value”, which quantifies statistical significance. We have changed the text to “p-value=0.92” in the text.
- Thanks, the superscript of 1 refers to “waiting spaces have the power to quell or exacerbate these feelings of anxiety”. We have moved the citation to the end of the sentence for clarity.
- That’s a great question. Unfortunately DOTT cannot track people or objects over time as it is designed to document information every 10 minutes. This is an area of future study.
- Thanks, this has been clarified in the Figure 2 legend.
- In cases where people are not sitting close to one another, it is possible to know whether people placed their items to the left or right. However, when people are sitting next to each other (eg seat 1 and 3), it is difficult to assess whether a personal item on seat 2 belongs to person in seat 1 or 3 due to limitations in DOTT. We have added this as a limitation in the discussion section.
- Yes, patients have access to free Wi-Fi. This has been added in the methods section.
- Thanks, this was completed.
Reviewer 2 Report
Dear author,
It was a pleasure reading the article.
clear, sound and well structured.
I feel as if it would be a good idea to show possible reorganisation potentials, or to show the best and the least good organisation patterns on the case study shown in the article. It would be usefull for the reader to see the actual direction for the space transformation. It is just an idea as to visually support valuable research results, in a way their immediate usability.
Author Response
Thanks, that is a good idea. We’ve added this as a figure in our discussion section.
Reviewer 3 Report
The presented work analyzes the use and preference of waiting space in an outpatient surgery clinic. As positive aspects, it is worth highlighting the evaluation of the use of these spaces and the value of the results to improve the future design of this type of space. Likewise, the graphic figures are interesting for understanding the work. Regarding the structure of the manuscript, it is correct and complete. The methodology is considered appropriate for the type of work presented, as well as the description of results.
As recommendations for improvement, the following are indicated:
- Describe the verbal consent given as an alternative to written consent and justify the reason for said decision.
- Describe the study population in greater detail. For example, define what type of patients they are in terms of the level of severity of the reason why they are waiting, etc... It is important to know the level of risk they are assuming in order to correctly evaluate the use of the room. Likewise, the definition of the average waiting time can help.
- Indicate more precisely the group of chairs that is defined as “chair facing the window”. Since the window is in a lateral position and not frontal. Regarding the decision of the choice of seat, it may also be relevant to indicate the type of view that is offered through that window.
- Determine the original source of the figures. Where appropriate, specifying that they are the authors' own elaboration.
- Add a graph showing the seat preference data according to the type of occupant (patient, companion...).
- It would be advisable to better detail the conclusions to which the analysis of the results lead. The conclusions are limited and do not clearly show the benefits of research.
Round 2
Reviewer 1 Report
Paper Review
Title: “Using Architectural Mapping to Understand Behavior and Space Utilization in a Surgical Waiting Room of a Safety Net Hospital”
Authors: Elizabeth N. Liao, Lara Z. Chehab, Michelle Ossmann, Benjamin Alpers, Devika Patel and Amanda Sammann
Manuscript ID: IJERPH–1960753
Journal: Int. J. Environ. Res. Public Health
The authors have addressed all of my comments. Therefore, I recommend “Accept and Publish”.
Best Regards,
Reviewer 3 Report
The improvements made are sufficient to recommend the publication of the work.
In the future, it is recommended to include the original source from which they are derived in the titles of graphs and tables (instead of indicating it in the text).